# Effects of ion composition on escape and morphology at Mars

Qi Zhang[1,2], Mats Holmström[1], and Xiao-Dong Wang[1]

[1]Swedish Institute of Space Physics, Kiruna, Sweden
[2]Department of Physics, Umeå University, Umeå, Sweden

**Correspondence:** Qi Zhang (qizhang@irf.se)

**Abstract.** We refine a recently presented method to estimate ion escape from non-magnetized planets and apply it to Mars. The method combines in-situ observations and a hybrid plasma model (ions as particles, electrons as a fluid). We use measurements from the Mars Atmosphere and Volatile Evolution (MAVEN) mission and Mars Express (MEX) for one orbit on 2015-03-01. Observed upstream solar wind conditions are used as input to the model. We then vary the total ionospheric ion upflux until the solution fits the observed bow shock location. This solution is a self-consistent approximation of the global Mars-solar wind interaction at the time of the bow shock crossing, for the given upstream conditions. We can then study global properties, such as the heavy ion escape rate. Here we investigate in a case study the effects on escape estimates of assumed ionospheric ion composition, solar wind alpha particle concentration and temperature, solar wind velocity aberration, and solar wind electron temperature. We also study the amount of escape in the ion plume and in the tail of the planet. Here we find that estimates of total heavy ion escape are not very sensitive to the composition of the heavy ions, or the amount and temperature of the solar wind alpha particles. We also find that velocity aberration has a minor influence on escape, but that it is sensitive to the solar wind electron temperature. The plume escape is found to contribute 29% of the total heavy ion escape, in agreement with observations. Heavier ions have a larger fraction of escape in the plume compared to the tail. We also find that the escape estimates scales inversely with the square root of the atomic mass of the escaping ion specie.

## 1  Introduction

Atmospheric escape is an important process in the Martian climate evolution (Jakosky et al., 2017). For present Mars, the dominant escape of atmospheric neutrals is through four channels: Jeans escape (Chaffin et al., 2017; Jakosky et al., 2018), photochemical reactions (Fox and Hać, 2009; Lillis et al., 2017), sputtering (Leblanc et al., 2018) and electron impact ionization (Zhang et al., 2020). Ions above the exobase get accelerated by the solar wind electric field and can escape. Measurements from Phobos 2 (Lundin et al., 1989), Mars Express (MEX) (Barabash et al., 2007; Nilsson et al., 2021) and Mars Atmosphere and Volatile Evolution (MAVEN) (Dong et al., 2017) and estimates from magnetohydrodynamic (MHD) (Ma and nagy, 2007; Regoli et al., 2018) and hybrid (Ledvina et al., 2017) models indicate that the heavy ions escape rate on Mars is between $10^{23} - 10^{25} s^{-1}$. The parameters affecting the solar wind interaction with the Martian atmosphere have been investigated, including the upstream conditions like extreme ultraviolet (EUV) radiation and solar wind dynamic pressure (Dong et al., 2017; Nilsson et al., 2021; Ramstad and Barabash, 2021), and crustal magnetic fields (Fang et al., 2015; Ramstad et al., 2016; Weber et al., 2021).

Not only the number of the escaping ions is of interest, but also the composition and morphology contribute to the understanding of ion escape. Observations by MEX and MAVEN has identified $O^+$, $O_2^+$ and $CO_2^+$ as the major escaping species (Carlsson et al., 2006; Rojas et al., 2018; Inui et al., 2019). These are also the species commonly used when modeling the interaction between Mars and the solar wind, using MHD or hybrid models (Harnett and Winglee, 2006; Ma et al., 2019). The morphology of the escaping ions has been observed to follow two broad pathways (Dong et al., 2015, 2017; Dubinin et al., 2017; Nilsson et al., 2021). The solar wind convective electric field accelerates ionospheric ions into what is usually denoted as the ion plume at Mars. There is also a more fluid-like escape of ions into the tail region behind the planet. These two pathways has also been seen both in observations (Dong et al., 2017; Nilsson et al., 2021) and in models (Dong et al., 2014; Holmström and Wang, 2015; Regoli et al., 2018; Ma et al., 2019; Brecht et al., 2017).

Both measurements and models have limitations when applied to study the escape of ionospheric ions. For the detection by instruments on the spacecraft, it's difficult to cover all energies, especially low energies, and the full $4\pi$ sr field of view. Furthermore, an in-situ observation is only at a certain place and time. To cover all of the interaction region, we need to accumulate data for a long time and rely on statistics. Therefore, observing the complete interaction region at a specific time is impossible with a single spacecraft. Using simulations, we can get a full three-dimensional picture at any instance. Nevertheless, the atmosphere are highly dynamic and it's impossible to include all relevant physics in the models. Therefore, we here use a recently proposed method to take advantage of both measurements and models, to get a global coverage of data and to enable detailed studies of physical processes.

We use the amount of mass-loading of the solar wind as a free parameter to combine the model and observations. Mass-loading of the solar wind flow occurs wherever thermal ions are inserted into the flow. Mass loading by planetary ions slows down the solar wind and raises the bow shock (Alexander and Russell, 1985; Vignes et al., 2002; Mazelle et al., 2004; Hall et al., 2016). Given similar upstream conditions, the standoff distance of the bow shock from the planet will depend on the degree of mass-loading, which is dependent on the amount of ions in the upper parts of the ionosphere. At Mars, heavy ions at the top of the ionosphere will provide the mass-loading, and wave-particle interactions will generate a bow shock in the colissionless solar wind plasma upstream of the planet (Szegö et al., 2000). We use observed upstream solar wind parameters as input for a hybrid plasma model, where the total ion upflux at the exobase is a free parameter. We then vary this ion upflux to find the best fit for the observed bow shock location. The method proposed has a very simplified ionospheric model. The reason for this simplified model is that we then have one free parameter that we can optimize to find the value that best fit the observations (of the bow shock location). We think that having such a simplified representation of the ionosphere is justified in view of the large spatial and temporal variations that have been observed (Chaufray et al., 2015; Fowler et al., 2022; Leelavathi et al., 2023). A more complicated ionospheric model that is fixed will have problems capturing these variations.

The method was introduced by Holmström (2022). The model used in that work was simplified, where only protons were considered in the solar wind, and solar wind velocity aberration was not included, and only one heavy ion specie was implemented in the ionosphere. Here we use a three-species ionosphere ($O^+$, $O_2^+$, $CO_2^+$), a solar wind with alpha particles and velocity aberration. Using this improved model, we investigate the effects of including these parameters in the model, in particular ion composition, on the escape and morphology.

## 2 Model implementation

### 2.1 Model description

In a hybrid model, electrons are treated as a massless fluid and ions are treated as individual particles accelerated by the Lorentz force (Holmström, 2022). The electric field is given by

$$\boldsymbol{E} = \frac{1}{\rho_I}(-\boldsymbol{J}_I \times \boldsymbol{B} + \mu_0^{-1}(\nabla \times \boldsymbol{B}) \times \boldsymbol{B} - \nabla p_e) + \frac{\eta}{\mu_0}\nabla \times \boldsymbol{B}, \tag{1}$$

where $\boldsymbol{B}$ is the magnetic field, $\rho_I$ is the ion charge density, $\boldsymbol{J}_I$ is ion current density, $p_e$ is the electron pressure, $\eta$ is the resistivity and $\mu_0$ is the vacuum permeability, respectively. Faraday's law is used to advance the magnetic field in time by

$$\frac{\partial \boldsymbol{B}}{\partial t} = -\nabla \times \boldsymbol{E} \tag{2}$$

We use Mars Solar Orbital (MSO) coordinates, where the origin is at the center of the planet, the $X_{MSO}$-axis is directed to the sun, the $Y_{MSO}$-axis is in the orbital plane, perpendicular to the $X_{MSO}$-axis, and opposite to Mars' motion. Then $Z_{MSO}$-axis completes the right-handed coordinate system. Our simulation domain is -11000 km $\leq X_{MSO} \leq$ 10000 km, -34300 km $\leq Y_{MSO} \& Z_{MSO} \leq$ 34300 km and the cell size is $h =$350 km. The Mars model has a sphere centered at the origin with a radius of 3380 km, representing the solid planet. We have a spherical obstacle with a radius of 3550 km (the inner boundary of the simulation), representing the exobase at the altitude of 170 km. All ions inside the obstacle are removed from the simulation. The resistivity is $7 \cdot 10^5$ $\Omega$m in the solid planet. Outside the planet the resistivity is $5 \cdot 10^4$ $\Omega$m, in the ionosphere and the surrounding plasma. The vacuum regions are defined as the regions with a plasma density less than 1% of the solar wind density and the resistivity in vacuum regions is $10^6$ $\Omega$m. The number of macro particles per cell at the inflow boundary (the $+X_{MSO}$ side of the simulation box) is 8 for protons, and 2 for alpha particles. The weight (number of real particles represented by one macro particle) of the ionospheric ion macro particles are set to the same weight as for protons. The time step, $\Delta t$, is 0.2 s. The heavy ions are produced on the dayside, drawn from a Maxwellian distribution with a temperature of 200 K. The exobase ion upflux decays from the subsolar point to the terminator by the cosine of solar zenith angle (Holmström and Wang, 2015). Each produced heavy ion is then moved radially outward by a distance randomly drawn from $[0, h]$. We run the model until a steady state is reached, after approximately 500 seconds of simulation time (when the number of heavy ions in the simulation domain remains on average constant). The escape rate is evaluated in term of number of ions per second.

We apply observed upstream solar wind parameters (solar wind density, solar wind velocity, and solar wind proton temperature from Solar Wind Ion Analyzer (SWIA); solar wind electron temperature from the Solar Wind Electron Analyzer (SWEA); and IMF from the Magnetometer (MAG)) at the inflow boundary. To derive these parameters, we calculated their median values of the undisturbed solar wind with MAVEN Key Parameters file outside the nominal bow shock (Vignes et al., 2000). Then we run several simulations with different heavy ion upflux rates at the exobase. Next, we compare the simulation results with observations in magnetic field and the proton density, to find the simulation run that best fits the observed bow shock location. The space resolution of these observations is higher than of model. We can then derive an escape rate estimate from this best fit

run. The total escape rate is computed by averaging the outflow in the region $X_{MSO} < -1.5R_m$ over 500 s to 600 s simulation time with 30 s interval.

We do not include any neutral corona in the model. On one hand, the effect of the corona on the ion escape is found to be minor (Dong et al., 2015). On the other hand, the effect of the corona will in a way be captured by our model. The additional mass loading from photoionization of neutrals will expand the bow shock location. In our model this will be compensated by requiring an increase in the ion upflux at the inner boundary. Crustal fields are also missing in the model. The bow shock location has been found to depend on the location of the magnetic anomalies relative to the solar wind flow (Fang et al., 2015; Garnier et al., 2022). It is unclear if this is because the fields expand the bow shock or because the presence of the fields increase the ion escape. The latter may not require crustal fields in the model used in our algorithm, as the parameter that we vary is the amount of ions near Mars available to escape. If the crustal fields in a specific geometry enhance escape, this will be captured in the algorithm because the best fit bow shock will be further out and require larger upflow at the inner boundary. In contrast, if the crustal fields in a specific geometry depress escape, the bow shock will be closer to the planet. An investigation of the effect of crustal fields on escape using our methodology is a topic for future studies.

## 2.2 Model example

In this study, we apply our method to one reference orbit from 13:00 to 15:00 UTC on 1 March 2015. Table 1 displays the upstream solar wind conditions for this orbit from MAVEN observations. We run three simulations with three different total exobase upflux rates listed in Table 2. All the runs are with the same input upstream conditions listed in Table 1, and with 5% number density of alpha particles (same upstream temperature and velocity as for protons), and with an exobase upflux composition of 54% $O^+$, 39% $O_2^+$ and 7% $CO_2^+$, which will be discussed later in Section 3.1.2. The simulation results are then compared with MAVEN measurements of magnetic field, solar wind velocity and proton density in Fig 1.

It is an optimization process to find a simulation run that best matches observations. We select different upflux values and perform simulations until we find a good fit to the bow shock location. By "good fit" we mean that the difference between a simulation and the observed bow shock along the spacecraft trajectory is on the order of the simulation cell size. This can require many model runs. For this reference orbit, we compare three simulation runs that have bow shock locations close to the observed one. By visual inspection, the Upflux 2 simulation fits the observation best. Upflux 1 gives a bow shock too close to the planet, since the mass loading is too small, while Upflux 3 gives a bow shock too far away from the planet. We see a good agreement between the model and observations in the magnetosheath region (the grey area in Fig 1). While closer to the planet, below the Induced Magnetosphere Boundary (IMB), the model magnetic field is not increasing as much as in the observation, but we do not expect a perfect fit due to the simplified ionosphere we use, and the lack of crustal magnetic fields in our model. We also verify the fit for the Upflux 2 simulation using MEX Electron Spectrometer (ELS) observations of bow shock crossings in Fig 2. This supports that Upflux 2 is the best fitting simulation run. In the rest of this work, we use this simulation as a reference.

**Table 1.** Upstream solar wind parameters in MSO coordinates from 13:00 to 15:00 on 1 March 2015 estimated from MAVEN observations

| | |
|---|---|
| Density [cm$^{-3}$] | 2.4 |
| Velocity [km/s] | (-350, 45, 12) |
| Proton temperature [K] | $1.2\times10^5$ |
| Electron temperature [K] | $1.7\times10^5$ |
| Interplanetary magnetic field [nT] | (-1, -2.7, -1) |

**Table 2.** The total exobase ion upflux and resulting total escape rates used for the three simulations

| Case | O$^+$ [s$^{-1}$] | O$_2^+$ [s$^{-1}$] | CO$_2^+$ [s$^{-1}$] | Escape rate [s$^{-1}$] |
|---|---|---|---|---|
| Upflux 1 | $4.6\times10^{24}$ | $3.2\times10^{24}$ | $6.1\times10^{23}$ | $5.08\times10^{24}$ |
| Upflux 2 | $5.0\times10^{24}$ | $3.6\times10^{24}$ | $6.7\times10^{23}$ | $6.78\times10^{24}$ |
| Upflux 3 | $5.5\times10^{24}$ | $3.9\times10^{24}$ | $7.3\times10^{23}$ | $8.94\times10^{24}$ |

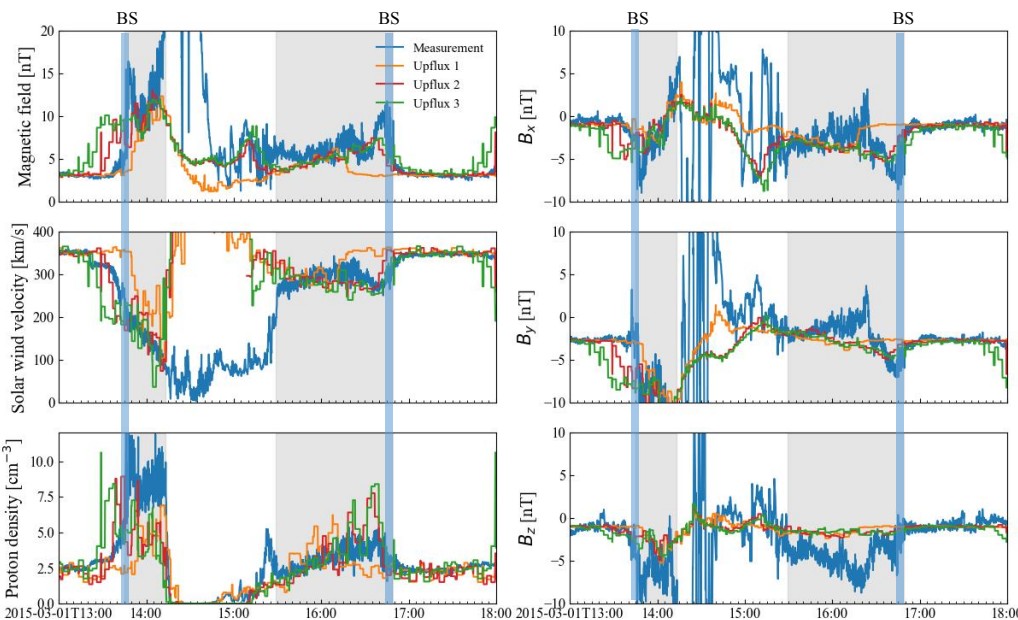

**Figure 1.** Model results compared to MAVEN measurements (blue lines). Orange, red and green lines are the simulation results for three different productions in Table 2, respectively. The left column shows a comparison for the magnetic field magnitude, the solar wind velocity and the proton density. The right column shows the three components of the magnetic field. The bow shock location is identified by the change in magnetic field and solar wind density. The blue areas indicate the bow shock locations. The grey areas indicate the magnetosheath regions.

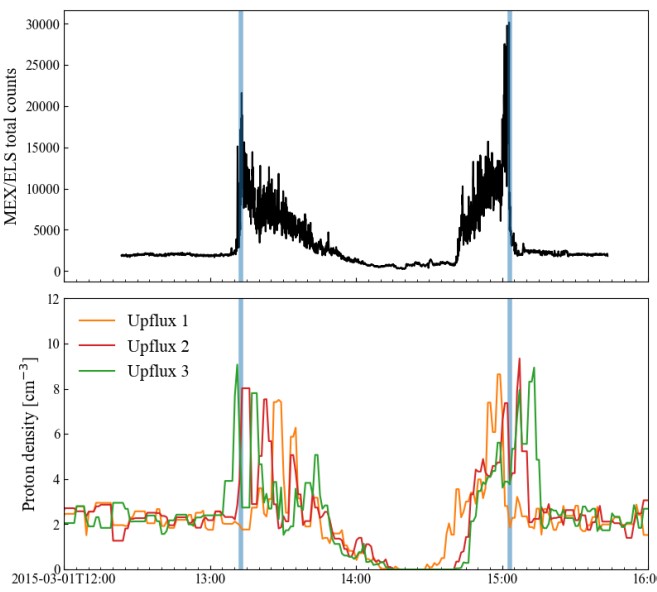

**Figure 2.** The top panel is MEX/ELS summed counts per scan, as a proxy of electron flux, to identify the bow shock location. The bottom panel is model results of Table 2 compared to MEX measurements. Blue lines are bow shock crossing times identified from MEX ELS observations.

## 3 Results and discussions

In what follows, we investigate the effects of alpha particles, heavy ion composition, velocity aberration, and solar wind electron temperature on escape estimates. We then study in what regions near the planet that ions escape.

### 3.1 Effects of parameters on escape estimates

#### 3.1.1 Effects of alpha particles on escape estimates

In addition to protons and electrons, the upstream solar wind contains a variable amount of alpha particles, $He^{++}$. Upstream solar wind observations by SWIA on MAVEN suggests a 3%-5% abundance of alpha particle populations in number density (Halekas et al., 2017b). Despite the low percentage, alpha particles carry up to ~20% of the solar wind kinetic energy due to its mass. Therefore, including alpha particles in the model will increase the kinetic energy density and dynamic pressure of the solar wind, thus impacting the solar wind interaction with Mars.

Furthermore, previous studies found that the alpha temperature is higher than the proton temperature and their ratio changes with heliocentric distance (Marsch et al., 1982; Von Steiger et al., 1995; Araneda et al., 2009; Hellinger and Trávníček, 2013) since the solar wind particles encounter parallel cooling and perpendicular heating driven by kinetic and Alfvén-cyclotron wave instabilities and heavier ions are preferentially heated (Araneda et al., 2009; Hellinger and Trávníček, 2013). At 1 AU, the ratio between alpha temperature ($T_\alpha$) and proton temperature ($T_p$) has been observed to vary from 2.5 to 5 (Bourouaine et

**Table 3.** Parameters used for the simulation runs investigating the effects of alpha particle, and the resulting escape estimates.

| Simulation Cases | Proton density [cm$^{-3}$] | Alpha density [cm$^{-3}$] | Alpha temperature | Escape rate [s$^{-1}$] |
|---|---|---|---|---|
| Case 1 | 2.4 | 0 | 0 | $6.44 \times 10^{24}$ |
| Case 2 (baseline) | 2.28 | 0.12 | $T_\alpha = T_p$ | $6.78 \times 10^{24}$ |
| Case 3 | 2.28 | 0.12 | $T_\alpha = 5T_p$ | $6.98 \times 10^{24}$ |

al., 2011; Wilson et al., 2018; Stansby et al., 2019).

We now investigate how our ionospheric heavy ion escape rate estimates are affected by alpha particle abundance and temperature in the solar wind. We ran two more cases in addition to the reference case with Upflux 2 in Section 2.2, which is Case 2, the baseline case, in this exercise. In Case 1, we only include protons. In Case 3, we increase the alpha temperature to 5 times

145 that in Case 2. We keep the total solar wind particle number density (sum of protons and alpha particles) the same. In Cases 2 and 3, the upstream solar wind contains 5% alpha particles. All the relevant parameters are listed in Table 3. The total exobase ion upflux used in these cases can be found in Table 4.

We find that the model escape rate estimate is slightly higher when we include alpha particles in the upstream solar wind (Case 2 in Table 3) than when we exclude them (Case 1 in Table 3). This is probably because including the heavier alpha particles

increases the dynamic pressure of the upstream solar wind, since we keep the total number density of the solar wind constant. The increased dynamic pressure requires a larger exobase upflux to keep the bow shock at the same location, hence increasing the escape.

To examine the effect of alpha temperature on ion escape in our method, we used $T_\alpha = 5T_p$ for comparison with the case of identical temperature for protons and alpha particles (Case 2 in Table 3). Hotter alpha particles (Case 3 in Table 3) with larger

thermal pressure compress the bow shock more, which requires an increased mass loading (from larger inner boundary heavy ion upflux). This finally leads to 3% more escape. Considering this small increase and the lack of knowledge of actual alpha temperature around Mars, later in this study, we keep the alpha particles temperature the same as for protons.

### 3.1.2 Effects of heavy ion composition on escape estimates

The heavy ion composition of the upper parts of an ionosphere directly influences the composition of escaping plasma, as well

as the dynamics of the escaping plasma due to their different mass per charge ratios. It is therefore important what composition of different ion species that we use in our model's exobase ion upflux. Carlsson et al. (2006) found that $O^+$ is the most abundant escaping species. They measured a flux ratio of $O^+/O_2^+/CO_2^+ = 10:9:2$ inside the IMB using MEX Ion Mass Analyser (IMA) nightside data. With the same instrument, Rojas et al. (2018) found a number ratio of $O^+/O_2^+ = 1.3$ averaged over the whole space inside the IMB. Inui et al. (2019) discovered larger $O_2^+$ flux than $O^+$ in the wake region based on MAVEN observations.

In summary, the measurements show uncertainties in the composition of escaping ions. Therefore it's of interest to explore the influence of different heavy ion compositions on escape estimates. In our model, we specify the ratio between the different

**Table 4.** The total exobase ion upflux [s$^{-1}$] and escape rate [s$^{-1}$] for all simulation runs.

| | $O^+$ | $O_2^+$ | $CO_2^+$ | $O^+$ escape rate | $O_2^+$ escape rate | $CO_2^+$ escape rate | Total escape rate |
|---|---|---|---|---|---|---|---|
| Case 1 | $4.6\times10^{24}$ | $3.2\times10^{24}$ | $6.1\times10^{23}$ | $3.24\times10^{24}$ | $2.69\times10^{24}$ | $5.1\times10^{23}$ | $6.44\times10^{24}$ |
| Case 2 | $5.0\times10^{24}$ | $3.6\times10^{24}$ | $6.7\times10^{23}$ | $3.46\times10^{24}$ | $2.77\times10^{24}$ | $5.5\times10^{23}$ | $6.78\times10^{24}$ |
| Case 3 | $5.1\times10^{24}$ | $3.6\times10^{24}$ | $6.8\times10^{23}$ | $3.56\times10^{24}$ | $2.86\times10^{24}$ | $5.6\times10^{23}$ | $6.98\times10^{24}$ |
| Case 4 | $5.0\times10^{24}$ | $3.6\times10^{24}$ | $6.7\times10^{23}$ | $3.69\times10^{24}$ | $3.03\times10^{24}$ | $5.5\times10^{23}$ | $7.27\times10^{24}$ |
| Case 5 | $5.0\times10^{24}$ | $3.6\times10^{24}$ | $6.7\times10^{23}$ | $3.54\times10^{24}$ | $2.80\times10^{24}$ | $5.4\times10^{23}$ | $6.88\times10^{24}$ |
| Case 6 | $5.0\times10^{24}$ | $3.6\times10^{24}$ | $6.7\times10^{23}$ | $3.46\times10^{24}$ | $2.77\times10^{24}$ | $5.5\times10^{23}$ | $6.78\times10^{24}$ |
| 0% $O^+$ | 0 | $7.8\times10^{24}$ | 0 | 0 | $5.79\times10^{24}$ | 0 | $5.79\times10^{24}$ |
| 26% $O^+$ | $2.2\times10^{24}$ | $6.2\times10^{24}$ | 0 | $1.42\times10^{24}$ | $4.39\times10^{24}$ | 0 | $5.81\times10^{24}$ |
| 48% $O^+$ | $4.4\times10^{24}$ | $4.7\times10^{24}$ | 0 | $2.97\times10^{24}$ | $3.67\times10^{24}$ | 0 | $6.64\times10^{24}$ |
| 58% $O^+$ | $5.5\times10^{24}$ | $3.9\times10^{24}$ | 0 | $3.83\times10^{24}$ | $2.99\times10^{24}$ | 0 | $6.82\times10^{24}$ |
| 68% $O^+$ | $6.6\times10^{24}$ | $3.1\times10^{24}$ | 0 | $4.62\times10^{24}$ | $2.37\times10^{24}$ | 0 | $6.99\times10^{24}$ |
| 85% $O^+$ | $8.8\times10^{24}$ | $1.6\times10^{24}$ | 0 | $5.96\times10^{24}$ | $1.19\times10^{24}$ | 0 | $7.15\times10^{24}$ |
| 100% $O^+$ | $1.1\times10^{25}$ | 0 | 0 | $8.37\times10^{24}$ | 0 | 0 | $8.37\times10^{24}$ |
| 100% $CO_2^+$ | 0 | 0 | $6.6\times10^{24}$ | 0 | 0 | $5.28\times10^{24}$ | $5.28\times10^{24}$ |

species, in addition to the total upflux, to fit the bow shock location. Here we consider $O^+$, $O_2^+$, and $CO_2^+$ since those are the major observed ion species, and the ones typically considered in models of the interaction between Mars and the solar wind (Kallio et al., 2008; Ma et al., 2014; Holmström and Wang, 2015). The total exobase upflux is computed by $nR_0^2\sqrt{\pi kT/2m_i}$, where n is the subsolar exobase density, $R_0$ the radius of the exobase, $k$ is Boltzmann constant, $T$ is the temperature at the exobase and $m_i$ is the mass of the ion specie.

In Fig 3, we present escape rates of seven different $O^+/O_2^+$ ratios of the total exobase upflux. We examine the $O^+/O_2^+$ ratio because $O^+$ and $O_2^+$ are the most abundant heavy ion species at Mars (Carlsson et al., 2006; Rojas et al., 2018; Inui et al., 2019). The total exobase ion upflux, and composition, used in all simulation runs in this paper can be found in Table 4. In every case, we adjust the total upflux rate to fit the observed bow shock location. As the $O^+$ upflux fraction increases from 0 to 100%, the total escape rate increases by 45%. So the escape is not inversely proportional to the mass of the escaping ion specie. In that case we would have an increase of 100%. Instead, the ratio of the escape for the cases of only $O^+$ exobase upflux and only $O_2^+$ upflux is close to $\sqrt{2}\approx1.41$, suggesting that escape scales inversely with the square root of the atomic mass of the escaping ion specie. To test this hypothesis, we made an unrealistic run with only $CO_2^+$ exobase upflux, resulting in an escape estimate that is 58% smaller than the $O^+$ case. This value is close to inverse of the square root of the mass ratio, $\sqrt{44/16}\approx1.66$, supporting the scaling hypothesis. It is not surprising that we do not have a perfect scaling when comparing the escape rates of the different species, since the escape process should not only have a dependence on the mass of different species, but will also be affected by the different trajectories due to differences in the mass per charge of the species.

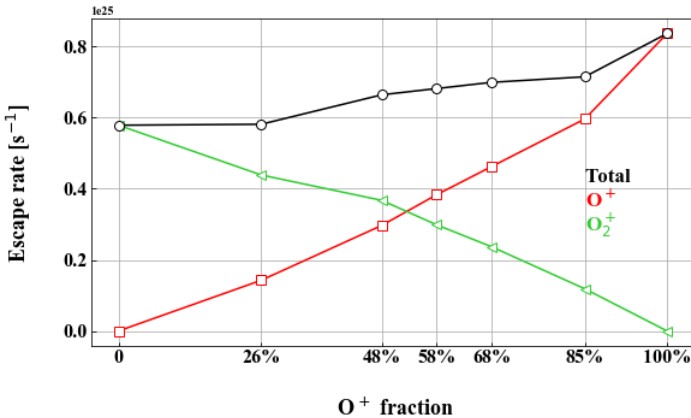

**Figure 3.** Escape rate for different compositions of the ion upflux at the exobase (inner boundary). The red line represents $O^+$ escape rate. The green line represents $O_2^+$ escape rate. The black line represents total escape rate. The $O^+$ fraction is of the total exobase number upflux, where the rest is $O_2^+$ in this experiment.

We can here only speculate as to why escape rates would scale inversely with the square root of the atomic mass of the escaping ion specie. Since flux is proportional to velocity, that in turn is proportional to the square root of kinetic energy divided by mass, we would have an inverse dependence on square root of mass, assuming that the kinetic energy is constant. So maybe the total energy flux of the escaping ions is similar, independent of the species of the escaping ions. This would mean that the same power is transfered from the upstream solar wind to the escaping ions. It also seems reasonable that the same power is required to keep the bow shock at the same distance.

We can note that there is only a 5% increase in escape as $O^+$ increases from 48% to 68%, indicating that the escape estimate is not so sensitive to the exact $O^+/O_2^+$ ratio of the exobase upflux. Therefore, we use 54% $O^+$, 39% $O_2^+$ and 7% $CO_2^+$ as the composition of the exobase upflux hereafter. This proportion comes from our selected subsolar exobase density fractions of 45% $O^+$, 45% $O_2^+$ and 10% $CO_2^+$.

### 3.1.3   Effects of solar wind velocity aberration on escape estimates

Velocity aberration is the deviation of the upstream solar wind velocity direction from the anti-sunward direction ($-X_{MSO}$). It is due to the planet's orbital motion around the Sun, and disturbances in the solar wind. At Venus the orbital velocity is around 35 km/s (Lundin et al., 2011, 2013), and possibly causes $O^+$ flow asymmetry in the plasma tail (Lundin et al., 2011) and large-scale flow vortex (Lundin et al., 2013). At Mars, the typical aberration angle is approximately 5° and usually ignored (Halekas et al., 2017a) for the tenuous and less viscous atmosphere.

In our model, the solar wind aberration is included since we use the upstream solar wind proton velocity vector observed by MAVEN SWIA. However, it is of interest how much variations of this angle affect the Mars-solar wind interactions since it is not completely stable in the upstream solar wind, and accurate velocity vectors are not always available. The aberration

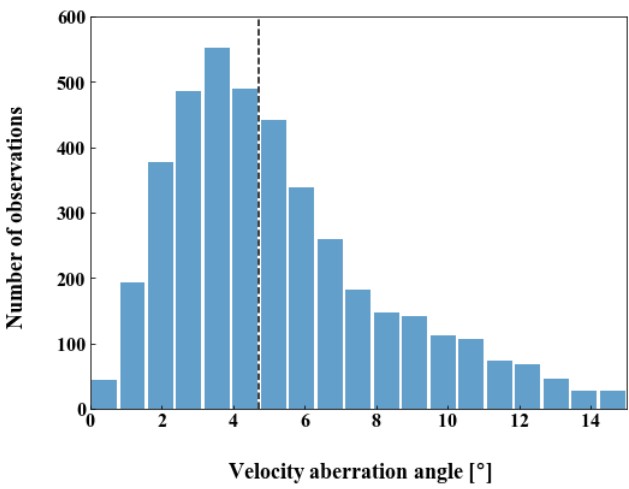

**Figure 4.** Velocity aberration distribution calculated from MAVEN SWIA solar wind observations during 2014-2019. The median velocity aberration angle ($4.7°$) is marked by a black dashed line.

introduces an asymmetry since the ionosphere in the model is still symmetric around the $X_{MSO}$-axis. Here we investigate the effects of aberration on the heavy ion escape and on the global solar wind interaction.

MAVEN can observe the full solar wind velocity vector. Examining 4117 orbits of SWIA data from November 2014 to November 2019, we compute a velocity aberration distribution (Fig 4). The median velocity aberration angle is $4.7°$. In some cases, it is up to $15°$. The solar wind velocity aberration angle of the orbit we used in this paper is $7.6°$. For comparison, we run two more simulations in addition to the baseline case (Case 6, the same as Case 2 with Upflux 2 in Section 2.2). In Case 4 we assume solar wind travels along $-X_{MSO}$ without velocity aberration. When we change the solar wind velocity direction, the

IMF cone angle (angle between solar wind velocity and the IMF) should rotate simultaneously to keep the same magnitude of the convective electric field. So in Case 5, with no velocity aberration, we have also rotated the IMF to keep the IMF cone angle the same as Case 6.

   In Table 5, we see the effects of different velocity aberrations on escape. When we assume no aberration (Case 4), the escape increases by 7% (compared to Case 6). When we rotate the upstream IMF to keep the cone angle the same (Case 5), the differ-

ence in escape (compared to Case 6) is only 1%. The conclusion is that the largest effect from different assumed aberrations is the different angles between the upstream solar wind velocity and the magnetic field, resulting in different upstream convective electric fields. The effect of having the upstream solar wind velocity at an angle relative to the ionosphere is minor in comparison.

   However, the effect of having the solar wind at an angle to the symmetry axis of the ionosphere (the $X_{MSO}$-axis) should be

larger at further distances in the tail behind the planet, since it represents a tilt of the whole induced magnetosphere. In Fig. 5 we examine this by looking at a plane at $X_{MSO} = -1.5R_m$ down the tail. We see that the morphology of the central heavy

**Table 5.** Parameters used for the simulation runs investigating the effects of velocity aberration, and the resulting escape estimates.

| Simulation Cases | Velocity [km/s] | IMF [nT] | Escape rate [s$^{-1}$] |
|---|---|---|---|
| Case 4 | (-353, 0, 0) | (-1, -2.7, -1) | $7.27\times10^{24}$ |
| Case 5 | (-353, 0, 0) | (-0.6, -2.1, -2.1) | $6.88\times10^{24}$ |
| Case 6 (baseline) | (-350, 45, 12) | (-1, -2.7, -1) | $6.78\times10^{24}$ |

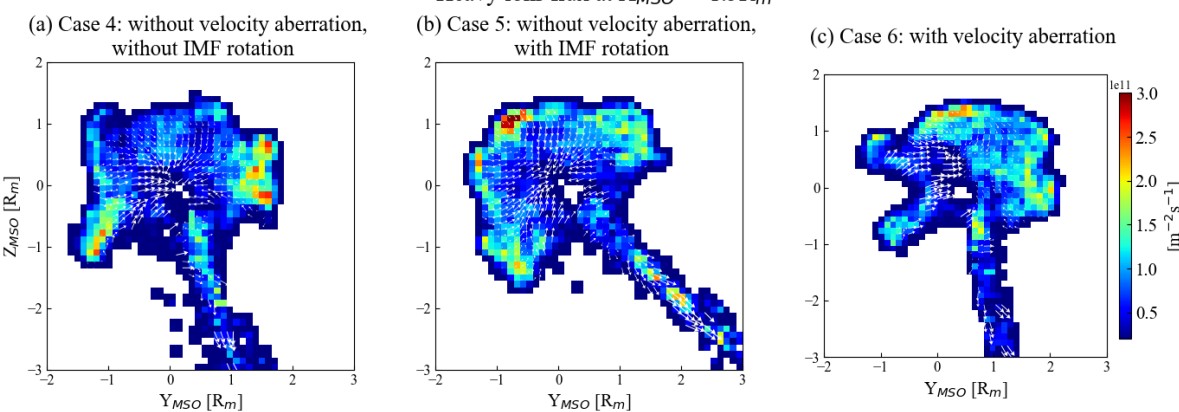

**Figure 5.** The panels show the heavy ions flux in $Y_{MSO} - Z_{MSO}$ plane sliced at $X_{MSO} = -1.5R_m$ of three cases described in Table 5. The white arrows denote the direction of flux. The length of arrows show the velocity magnitude, with larger velocities in the energetic plume region.

ion fluxes are quite different. The maximum flux is distributed in different regions in three cases. Case 4 has the maximum flux on both $+Y_{MSO}$ side and $-Y_{MSO}$ side. In case 5, large flux is widely distributed in the margin. While in Case 6, most of the flux is concentrated on $+Y_{MSO}$ side. The direction of the plume flux is also different. Small differences are magnified as we go further distances down the tail. The rotation of the magnetosphere is visible comparing Case 5 and 6, where the plume direction is visibly tilted.

In conclusion, the effect of velocity aberration is a tilt of the whole magnetosphere, so that the line of symmetry of the magnetosphere is along the upstream flow direction instead of the $X_{MSO}$-axis. This effect will be larger further behind the planet. If the angle between the upstream magnetic field and velocity remains the same, the effects on the interaction (except for the tilt) will be small. If however this angle change, it will affect the global interaction, probably due to the change in the upstream convective electric field.

### 3.1.4 Effects of electron temperature and ambipolar field on escape estimates

The ambipolar field plays a role in the solar wind interaction with Mars by enhancing ion loss in the collisionless ionosphere above the exobase (Ergun et al., 2016; Brecht et al., 2017; Ma et al., 2019). In the upper Martian atmosphere, electrons diffuse

faster than ions and an electric field is generated in the direction against the density gradient, called the ambipolar field. Ergun et al. (2016) showed increased $O_2^+$ outflow with increasing high-altitude electron temperature. Xu et al. (2021b) utilized electrostatic potential from MAVEN measurements (Xu et al., 2021a; Horaites et al., 2021) to estimate the global ambipolar field at Mars, which agrees well with MHD model predictions (Ma et al., 2019).

The ambipolar field cannot be self-consistently represented in MHD and hybrid models due to the assumptions of charge neutrality. There are different approaches of how to include the effects of the ambipolar electric field in the models. It is therefore of interest to look at how the ambipolar field is approximated in a hybrid model. The ambipolar field in our model is derived from the gradient of the electron pressure, $p_e = n_e k T_e$. Thus our ambipolar field is related to the electron temperature. In our model, the electron pressure is isotropic and computed from the charge density by (Holmström, 2010)

$$\frac{p_e}{p_{e0}} = \left(\frac{\rho_e}{\rho_{e0}}\right)^{\gamma} \tag{3}$$

where $\gamma = 5/3$ is the adiabatic index, $p_{e0}$ and $\rho_{e0}$ the electron pressure and electron charge density in the upstream solar wind, where $p_{e0} = n_{e0} k T_{e0}$. $k$ is the Boltzmann constant and $T_{e0}$ is the solar wind upstream electron temperature. Here we get

$$p_e = A\rho_e^{\gamma} \quad \text{with} \quad A = \frac{k}{m_e} T_{e0} \rho_{e0}^{1-\gamma} \tag{4}$$

where $\rho_e \equiv \rho_I$ and $\rho_{e0} \equiv \rho_{I0}$ for charge neutrality. Since the ambipolar term in Equation (1) is calculated from the negative gradient of the electron pressure, this electric field will be largest in regions where the total charge density has the largest gradient.

Figure 6 shows the electron pressure and the ambipolar electric field. We can note that the magnitude of the ambipolar field is largest at the bow shock and at the IMB. The magnitude of our ambipolar field is up to 0.1 V/km at the boundaries. At the topside of the Martian ionosphere (dark red region closed to planet in Fig 6), the ambipolar field energizes $O^+$, $O_2^+$ and $CO_2^+$ with accelerations of nearly 0.6, 0.3 and 0.2 km/s$^2$, which can lead to the heavy ions escaping (Kar et al., 1996). The black arrows in Fig 6 (d), (e), and (f) show the direction of the ambipolar field. The field points outwards at both boundaries. At the bow shock, the ambipolar field direction is consistent with MAVEN observations (Figure 2 in Xu et al. (2021b)) and MHD model results (Figure 3 in Xu et al. (2021b)). On the IMB, the ambipolar field in our model is directed outward while some observations suggest however that the field is directed inward in that region due to the electron pressure gradient from the colder ionosphere to the hotter magnetosheath (Xu et al., 2021b). The reason for this discrepancy might be that as is common in hybrid models we use an adiabatic approximation for the electron pressure term. This means that the resulting ambipolar field term will be directed opposite to charge density gradients. The electron temperature, and thereby the electron pressure, is a free parameter in hybrid models. Electron density and current is given from quasi-neutrality and Ohm's law, respectively. It would be of interest in future work to study in detail alternatives to the adiabatic approximation. One approach taken is to assume an electron temperature profile in the ionosphere (Bößwetter et al., 2004; Modolo et al., 2016). Another approach is to solve a fluid flow equation for the electrons (Brecht et al., 2017).

To test the sensitivity of the escape rate to changes in the upstream electron temperature, we run three cases with different upstream electron temperatures (the minimum, the median and the maximum temperature observed in the undisturbed solar

wind) but with other parameters the same. The results are shown in Figure 7. The observed solar wind electron temperature varies in the solar wind in the range given by the x-axis. This results in the variation in escape are shown on the y-axis. So the uncertainty in electron temperature gives an escape in the range $6.5 - 7.0 \times 10^{24}$. This is the uncertainty in escape caused by the electron temperature uncertainty. We see that the escape rate is sensitive to the assumed upstream electron temperature and increases with it, probably because the larger electron temperature leading to a larger ambipolar field accelerates more ions to escape energies. Since we use the observed upstream electron temperature in our model, this is not a problem, except for measurement uncertainties. However, it indicates that escape estimates are sensitive to the model assumptions regarding the ambipolar fields. The effects of different approaches to include the effects of charge separation in a hybrid model, and how it affects model results, would be a topic of future studies.

## 3.2 Morphology of heavy ions escape

Our method of combining observations and modeling gives us a self-consistent description of the Mars-solar wind interaction, which can be used to study other properties of the solar wind interaction than escape. We now examine the morphology of the escaping ions, using the exobase upflux ion composition of 54% $O^+$, 39% $O_2^+$ and 7% $CO_2^+$. In the upstream solar wind we have 5 % solar wind alpha particles, with the same temperature as protons, as discussed before.

At Mars the escaping ionospheric ions usually form two major outflow channels: A cold fluid-like outflow in the tail behind the planet, and a more energetic outflow in the direction of solar wind convective electric field (Holmström and Wang, 2015). The escaping ions accelerated by the convective electric field $-\mathbf{V}_{SW} \times \mathbf{B}$, are usually called the ion plume at Mars. The Martian ion plume has been observed by MAVEN (Dong et al., 2015, 2017; Dubinin et al., 2017) and MEX (Nilsson et al., 2021), and modeled by multifluid MHD (Dong et al., 2014; Regoli et al., 2018; Ma et al., 2019) and hybrid codes (Holmström and Wang, 2015; Brecht et al., 2017). It is a matter of definition how to separate tail and plume fluxes, in observations and models. Dong et al. (2017) separated plume and tail flux by energy (>1 keV ions belong to the plume) and found that plume escape contributes 30% to total escape in low EUV conditions and 20% in high EUV conditions. Nilsson et al. (2021) defined the escape morphology using geometric box and called the outflow perpendicular to the X-axis "radial escape". They found that the radial escape does not depend on the solar cycle, but that the highest radial escape occurs at highest solar wind dynamic pressure conditions, and that the radial escape is around 20% to 40% of the total escape. Previous studies show that the amount of plume and radial escape, as a fraction of total escape, is not very sensitive to the exact definition chosen.

To separate escaping ions into plume and tail, we define a three-dimensional box in the simulation domain with a size similar to Nilsson et al. (2021) and Dong et al. (2017). Our box is defined by $X_{MSO} = \pm 1.6$ R$_m$ and $Y_{MSO}, Z_{MSO} = \pm 1.7$ R$_m$. Using a +1.6 R$_m$ as the boundary in the $+X_{MSO}$ direction instead of +2 R$_m$ used in other studies (Dong et al., 2017; Nilsson et al., 2021) will not affect our results since there is little heavy ion flux beyond $X_{MSO} = +1.6$ R$_m$. We define the outward ion fluxes through the $\pm Y_{MSO}$ and $\pm Z_{MSO}$ sides of the box as the plume flux and the fluxes through the $-X_{MSO}$ side as the tail flux (Fig 8).

Fig 9 displays our model results of Case 2 in Table 3 for the flux of the three heavy species in three of the planes. The first row in Fig 9 displays the tail flux and the second and third rows display the plume flux. We obtain a plume escape rate of $1.96 \times 10^{24}$

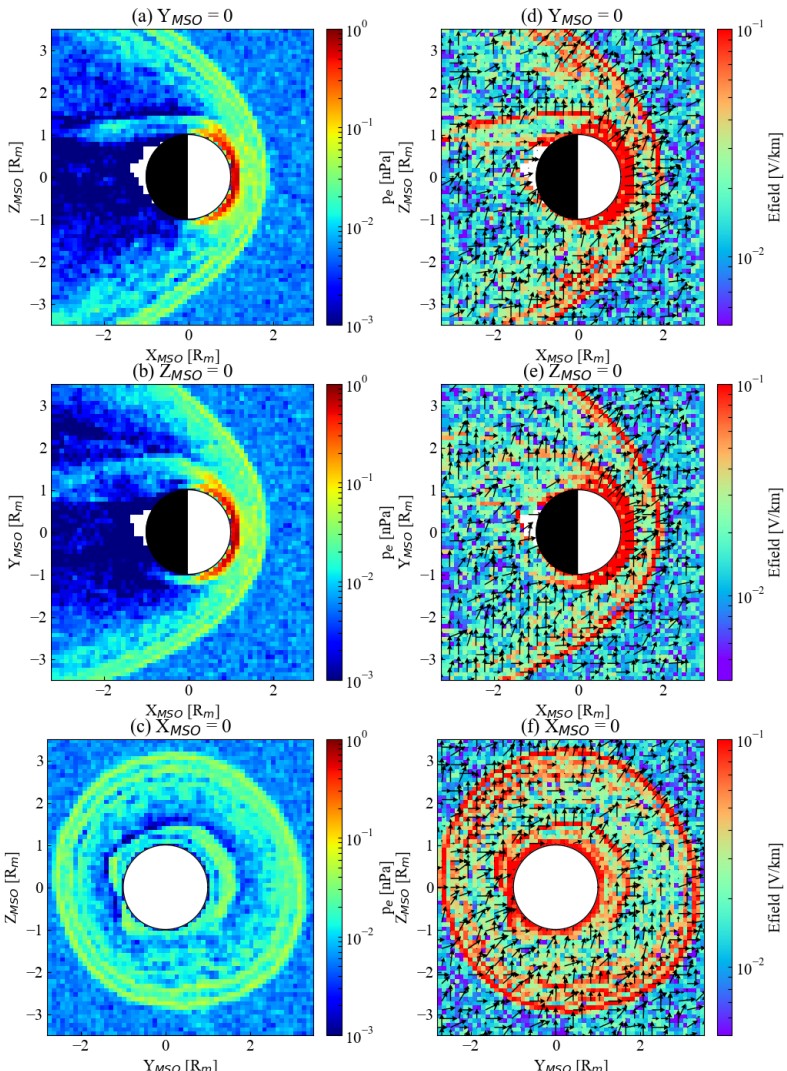

**Figure 6.** Electron pressure (a), (b), (c) and ambipolar electric field (d), (e), (f) from hybrid model. (a) and (d) is in $X_{MSO} - Z_{MSO}$ plane at $Y_{MSO}$=0. (b) and (e) is in $X_{MSO} - Y_{MSO}$ plane at $Z_{MSO}$=0. (c) and (f) is in $Y_{MSO} - Z_{MSO}$ plane at $X_{MSO}$=0. Black arrows in (d), (e) and (f) represent the direction of ambipolar electric field in each plane.

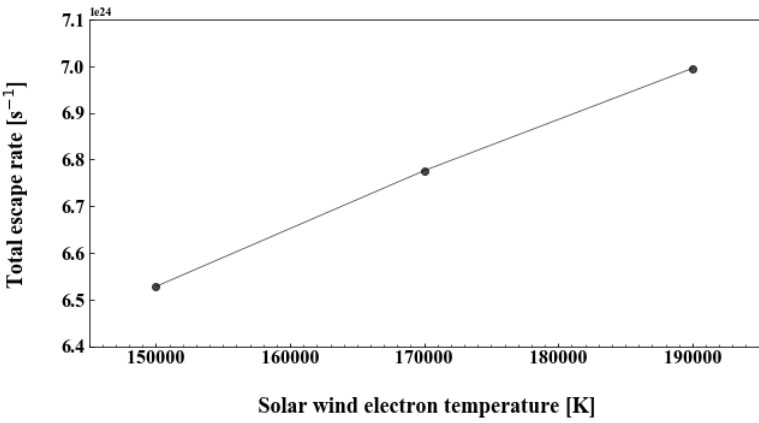

**Figure 7.** The total escape rate of three tests with various electron temperatures. The electron temperature in the middle ($1.7 \times 10^5$ K) is the median value of the undisturbed upstream solar wind and is what we used elsewhere in this study. $1.5 \times 10^5$ K and $1.9 \times 10^5$ K are the minimum and maximum values in the undisturbed solar wind from observations.

**Table 6.** The percentage for each ion specie in tail escape, plume escape and total escape, in terms of number of ions.

| Ion specie | Tail | Plume | Total |
|---|---|---|---|
| $O^+$ | 58% | 37% | 51% |
| $O_2^+$ | 36% | 51% | 41% |
| $CO_2^+$ | 6% | 12% | 8% |
| All | 100% | 100% | 100% |

$s^{-1}$, accounting for 29% of the total ion escape. This number is close to the observation results discussed above and a bit lower than MHD model results (35% to 45%) (Regoli et al., 2018). Table 6 illustrates that $O^+$ escape is dominant in the total and tail escape. $O_2^+$ is dominant in the plume escape. Higher $O_2^+$ composition in plume could be due to the fact that heavier ions, with larger gyro radius, easier to escape as plume since the flow is more perpendicular to the $X_{MSO}$-axis.

## 4    Conclusions

We have improved a new method for modeling the interaction between the solar wind and Mars, which uses a hybrid model to fit the observed bow shock location to determine a corresponding exobase ion upflux. The method was applied to one MAVEN orbit #811 on 2015-03-01, to investigate the effects on ion escape estimates of assumed heavy ion composition in the ionosphere, alpha particles in the solar wind, solar wind velocity aberration and electron temperature. We also studied ion escape rate in the plume and in the tail of the planet.

1. We find that ion compositions at the exobase with larger mass leads to a smaller estimate of the escape rate. The escape estimate is inversely proportional to the square root of the atomic mass of the escaping ion specie. However, the escape

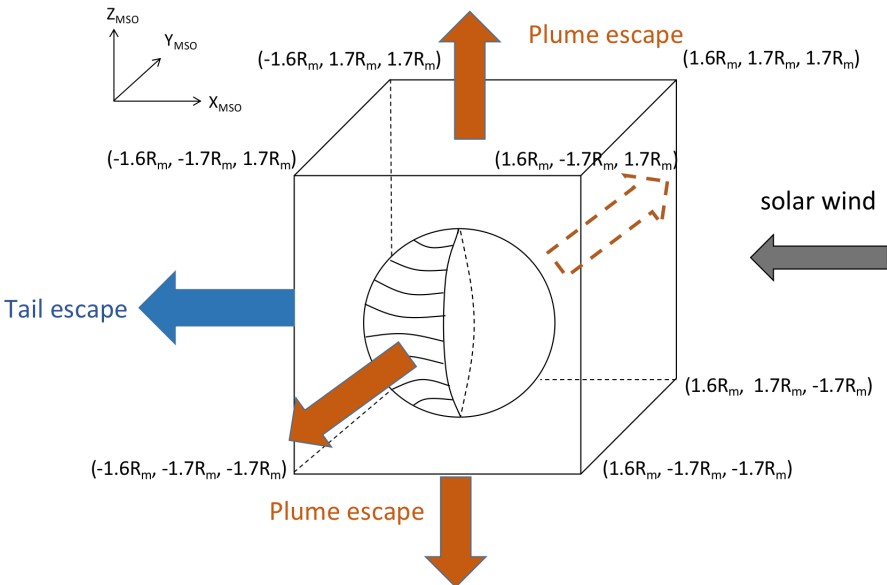

**Figure 8.** Illustration of how plume and tail escape is defined in this study. The flux passing through the $Y_{MSO} - Z_{MSO}$ side of the box along $-X_{MSO}$ we define as tail flux (the blue arrow). The flux passing through the $X_{MSO} - Z_{MSO}$ and $Y_{MSO} - Z_{MSO}$ sides of the box is defined as plume flux (the orange arrows). The direction of the convective electric field determine the direction of the plume flux.

does not change substantially as the mixing ratio of $O^+$ relative to $O_2^+$ varies between 0.4 and 0.6, the range of observed composition of heavy ion fluxes.

2. We also find that the assumed fraction, and temperature, of alpha particles in the upstream solar wind, have a small effect on escape estimates. The escape increase by 5% when having a number fraction of 5% alpha particles in the upstream solar wind. Adding alpha particles increases the mass density of the upstream solar wind, compressing the bow shock. We then need a larger mass loading from heavy ion upflux at the exobase, resulting in larger escape. This was the case when the temperature of the upstream protons and alpha particles were assumed to be equal. If we assume a five times alpha temperature, we see a further 3% increase in escape due to the higher thermal pressure in the upstream solar wind further compressing the bow shock.

3. The effect of solar wind aberration on escape rate is found to be 7%. This is however when only rotating the upstream solar wind velocity. If we also rotate the upstream magnetic field we find a change of only 1%. So the larger effect is from having a different angle (cone angle) between the solar wind velocity and the IMF. The smaller effect is from having the upstream solar wind impacting the ionosphere from a different direction than the anti-sunward direction.

4. We find that the escape rate is sensitive to the assumed upstream electron temperature and increases with it. This indicates a sensitivity to the model assumptions regarding the ambipolar fields. In our model we find ambipolar field strength at boundaries up to 0.1 V/km.

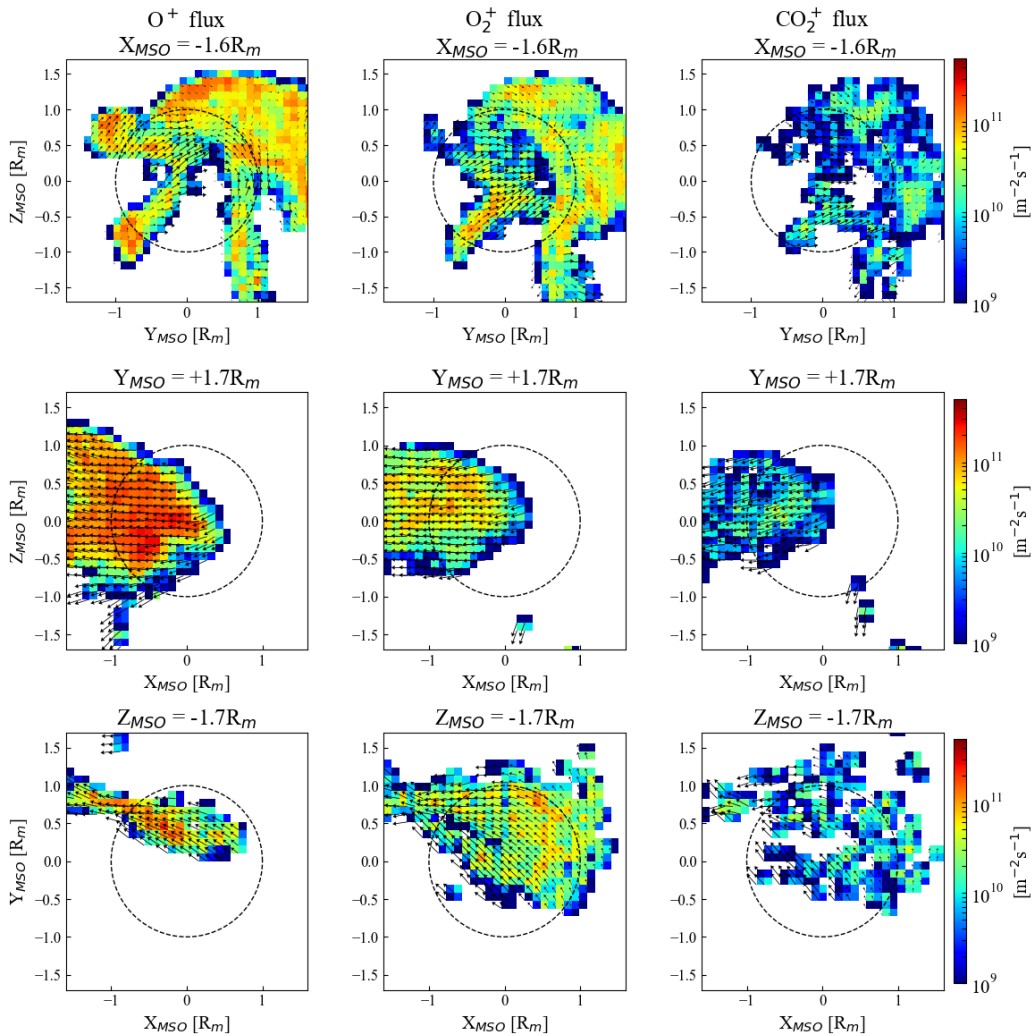

**Figure 9.** The distribution of the tail and plume flux divided according to Figure 8. The leftmost column shows $O^+$ flux, the middle column $O_2^+$ flux, and the rightmost column $CO_2^+$ flux. The first row show tail flux, and the two lower rows show plume flux. The black arrows indicate the direction of the flux. Notice that for this simulation there is no plume flux through the $-Y_{MSO}$ or $+Z_{MSO}$ sides.

5. We also studied the amount of escaping ions in the plume and the tail and find that 29% of the ions escape in the plume, consistent with observations. We also find that the fraction of ions, relative to the total escape, escaping in the plume increase with the mass of the ion specie. Possibly due to kinetic effects due to larger gyro radius.

This paper improves on our recent-proposed method and studies the role of some basic parameters on ion escape estimates at Mars. Future studies will further explore how upstream solar wind conditions and planetary conditions affect estimates of Martian heavy ion escape.

*Author contributions.* QZ initiated the study, run MH's models, and wrote the paper. All co-authors contributed to the discussion and the reviewing of the paper.

*Competing interests.* The authors declare that they have no competing interests.

*Acknowledgements.* This work was supported by The Swedish National Space Agency, Grant 198/19. Computing resources used in this work were provided by the Swedish National Infrastructure for Computing (SNIC) at the High Performance Computing Center North (HPC2N), Umeå University, Sweden. The software used in this work was in part developed by the DOE NNSA-ASC OASCR Flash Center at the University of Chicago.
The ASPERA-3 electron data used, from the ELS sensor, is available in ESA's Planetary Science Archive (PSA) at ftp://psa.esac.esa.int/pub/mirror/MARS-EXPRESS/ASPERA-3/MEX-M-ASPERA3-3-RDR-ELS-EXT5-V1.0/
The MAVEN data used in this work, ion data from the SWIA instrument and magnetic field data from the MAG instrument, is available in University of Colorado Boulder's Planetary Data System (PDS) at https://lasp.colorado.edu/maven/sdc/public/data/sci/kp/
We thank Yoshifumi Futaana for providing the ELS counts used in the paper.

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
