# Peer review of "Effects of ion composition on escape and morphology at Mars"

_EGUsphere, 2023_

## Author Response (AR1)

We thank the reviewer for insightful and constructive comments that we think will improve the paper. Please find our replies below to some of the issues brought up. The content of these replies would be incorporated in a revised version of the paper.

This work assumes an ionospheric ion composition that is not well explained. Section 2.2. says "it will be discussed later" but I do not think there is a real discussion later. I would suggest to proper justify it in this section with references.

We agree that the reference and description was not clear. The discussion about the composition we selected is in section 3.1.2, along with the justification for the selected composition. After examining the escape for different compositions, we conclude at the end of 3.1.2 that the escape is not very sensitive to the exact composition. Since some observations indicate more O+ and some more O2+, we selected equal O+ and O2+ densities at the exobase, with also 10% of CO2+.

The resulting upflux is then calculated by $nR_0^2 \sqrt{(\pi kT/2m_i)}$, where n is the subsolar exobase density, $R_0$ the radius of the exobase, k is Boltzmann constant, T is the temperature at the exobase and $m_i$ is the mass of the ion specie. Our selected density fractions of 45% O+, 45% O2+ and 10% CO2+ then results in exobase upflux proportions of 54% O+, 39% O2+ and 7% CO2+.

We will incorporate this explanation in the revised paper.

Also in Section 2.2, it would be good to explain a bit more the different Upflux scenarios of Table 2, which are very briefly mentioned, but a reader may not understand why these 3 scenarios are done, why are they important or actually their main differences.

This is an optimization process to find a simulation run that best match observations. We select different upflux values and perform simulations until we find a good fit to the bow shock location. By "good fit" we mean that the difference between a simulation and the observed bow shock along the spacecraft trajectory is on the order of the simulation cell size. This can require many model runs. In the paper we present the best fit (Upflux 2) along with the two bracketing runs that give a bow shock location that is too far or too close to the planet (Upflux 1 and 3).

This description will be added in the revised paper.

Lines 49-51: It is very hard to understand the meaning of this sentence. Could it please be rephrased?

Yes, that explanation was maybe too brief.

We will replace it by "We can note that the method proposed has a very simplified ionospheric model, where the only free parameter is the upflux of ionospheric ions at the subsolar point. The reason for this simplified model is that we then have one free parameter that we can optimize to find the value that best fit the observations (of the bow shock location).    We think that having such a simplified representation of the ionosphere is justified in view of the large spatial and temporal variations that has been observed (Chaufray, 2015; Fowler, 2022; Leelavathi, 2023). A more complicated ionospheric model that is fixed will have problems capturing these variations.''

The new description will be added in the revised paper.

Line 68: Why X_mso is not symmetric?

Actually it is not important if the extent of the simulation domain along each axis is symmetric or not, as long as the simulation domain is large enough to contain all of the interaction region. So the simulation domain is selected to contain the bow shock.

Lines 70 and 71: two different quantities for the radius of the planet are given. Please, unify them.

The Mars model has a sphere centered at the origin with a radius of 3380 km, representing the solid planet. We have a spherical obstacle with a radius of 3550 km (the inner boundary of the simulation), representing the exobase at the altitude of 170 km. All ions inside the obstacle are removed from the simulation. The resistivity is 7e5 Ωm in the solid planet. Outside the planet the resistivity is 5e4 Ωm, in the ionosphere and the surrounding plasma.

This new description will be added in the paper.

I believe a discussion section is needed. As any model, several upgrades are still needed to incorporate, such as the role of crustal fields at Mars or the neutral corona. Some discussion about these two aspects should appear in the paper with relation to what we know from empirical experience and other models, and how this model compares to others. In addition, it would be good to discuss how the numbers/results provided in this work would compare with other observations at different solar activity levels, seasons, etc. Future work may be needed to get a comprehensive picture, but based on the literature, I believe it is necessary to understand how this model may potentially compared under different circumstances, or what to expect from the model.

We have the combined Section '3 Results and discussions'.

The corona was explained in section 2.1. ' We do not include any neutral corona in the model. We can however note that in our method, additional mass loading from photoionization of neutrals in the exosphere will be compensated for by larger heavy ion upflux at the exobase, to achieve the observed bow shock location.'

We should discuss more about crustal fields in the paper. The bow shock location has been found to depend on the location of the magnetic anomalies relative to the solar wind flow (Fang et al., 2017; Garnier et al., 2022). It is unclear if this is because the fields expand the bow shock or because the presence of the fields increase the ion escape. The latter may not require crustal fields in the model used in our algorithm, as the parameter that we vary is the amount of ions near Mars available to escape. If the crustal fields in a specific geometry enhance escape, this will be captured in the algorithm because the best fit bow shock will be further out and require larger upflow at the inner boundary. In contrast, if the crustal fields in a specific geometry depress escape, the bow shock will be closer to the planet. An investigation of the effect of crustal fields on escape using our methodology is a topic for future studies.

This discussion about crustal fields will be added to the paper.

It's difficult to compare our results with other models because the methodologies are so different. We do not assume an ionosphere, as is usually done, but instead fit the model to observations.

We have submitted a follow-up paper to JGR, about how solar activity and upstream solar wind conditions affect the ion escape using our method. So in this paper we do not discuss that, instead focusing on some basic parameters that have been rarely investigated before.

Data: MAVEN key parameters are used. However, there are tons of them! It would be much appreciated a proper description of the parameters used and from which instrument they come from, if they have any caveats, etc. In addition, in line 179, the 4117 orbits examined, are also key parameters? Please describe it.

It includes solar wind density, solar wind velocity, and solar wind proton temperature from SWIA; solar wind electron temperature from the Solar Wind Electron Analyzer (SWEA); and IMF from the Magnetometer (MAG).

Yes, the 3D solar wind velocity in 4117 orbits is also computed from SWIA H+ solar wind velocities in the kp file.

 This description will be added in the paper.

Figure 1: it would be good to indicate the boundaries with a small label at the top of each panel.

That is a good suggestion. We will add such labels.

Figures 5 and 9: the colorbar does not have labels and units.

The label and unit are in the title of the plot. I will change that in the revised paper since it's not easy to find.

Line 245 and Figure 7: Text talk about uncertainties but figure does not show them. Can the uncertainties be included in the figure? A figure of this kind should have errorbars.

This was not clearly explained. Figure 7 actually shows the uncertainties, but not with error bars. The observed solar wind electron temperature varies in the solar wind in the range given by the x-axis. This results in the variation in escape are shown on the y-axis. So the uncertainty in electron temperature gives an escape in the range 6.5-7.0e24. This is the uncertainty in escape caused by the electron temperature uncertainty.

We will include this discussion at the end of 3.1.4.

Line 273: Do authors mean of the "total ion escape"? Or also including the neutral?

Total ion escape

 This will be changed in the paper.

Other comments:

The suggestions below will be implemented in a revised paper.

Line 5: "at this moment". Do you mean "in Mars' today"?

We will change it to "at the time of the bow shock crossing"

Line 20: it would be good to spell out MEX and MAVEN

Line 26: Observations "have". Also in this line, from which spacecraft are these "observations"?

Observations by MEX and MAVEN.

Line 24: spacecraft (no "s")

Line 54: "and" a solar wind...

Line 77: remove one "the"

Line 120: "Outflow 2", do authors mean "Upflux 2"?

Yes

Line 225: "The ambipolar field is largest at two boundaries, the bow shock and, closer to the planet, the IMB. " Can authors rephrase this sentence? Grammar is a bit difficult to follow.
''We can note that the magnitude of the ambipolar field is largest at the bow shock and at the IMB."
Lines 295-296: replace "major" and "minor" with "larger" and "smaller".

############################################################################

> A general comment on the study is that by considering the number of escaping ions, it is not clear that the escape of diatomic particles produces a double loss of total oxygen content. I think that this is an important consideration that reserves some discussion along the paper. So when the authors state that the escape is less, they should also evaluate the total number of oxygen atom loss, that is the most important parameter when considering evolution scenarios.
We agree that to estimate atomic oxygen escape is helpful to understand water loss. However, at present, O escape is dominant by neutral escape. O neutral escape is 1-2 order of magnitude larger than ion escape. If we want to include O atom escape discussion, we can not ignore neutral escape evaluation. But in this paper, we select to only focus on the ion escape and do not mention any neutral escape process. Also in the section 3.1.2, we discussed why escape is different for the three ionospheric ion species considered in this paper, and concluded that it is due to kinetic effects. We can also note that it is possible to compute the number of escaping oxygen atoms from Table A.1. by simply multiplying O2+ and CO2+ escape by two and adding the escape for the different species.

> Detailed comments:
> abstract: in the abstract it is not clear that the study refers to a single case. I suggest to add in line 7 "Here we investigate in a case study the effects…."
This suggestion will be implemented in a revised paper.

> Line 30: the definition of plume is given in section 3.2. I think that it should be given here.
We will expand the definition on Line 30 to "The solar wind convective electric field accelerates ionospheric ions into what is usually denoted as the ion plume at Mars."

> Lines 69-70: it is not totally clear that the expression -34300 km <YMSO,ZMSO  < 34300 km is a single one since a comma also is separating the previous expression. Maybe an & symbol can be used.
It will be revised as '-34300 km < $Y_{MSO}$ & $Z_{MSO}$ < 34300 km'.

> Lines 80-82: could the author better explain this point?
We do not include any neutral corona in the model. On one hand, the effect of corona on the ion escape is found to be minor (Dong15). On the other hand, the effect of corona will in a way be captured by our model. The additional mass loading from photoionization of neutrals will expand the bow shock location. In our model this will be compensated by requiring an increase in the ion upflux at the inner boundary.
This new description will be added in the paper.

> Line 88 and table 2: it must be specified that the escape rate is evaluated in term of number of ions per second.

We will add the unit specification in the revised paper.

> Figure 1: it is not very clear that the red lines fit better the observations. If the most relevant parameter that should be considered are the boundaries, this should be better explained in the text.

We agree that the description was not so clear. Another reviewer has the same question.

This is an optimization process to find a simulation run that best match observations. We select different upflux values and perform simulations until we find a good fit to the bow shock location. By "good fit" we mean that the difference between a simulation and the observed bow shock along the spacecraft trajectory is on the order of the simulation cell size. This can require many model runs. In the paper we present the best fit (Upflux 2) along with the two bracketing runs that give a bow shock location that is too far or too close to the planet (Upflux 1 and 3).

This description will be added in the revised paper.

> Figure 2: I cannot see MEX/ELS observations. The electron data from which the boundaries are derived should be shown in the plot or in a separate plot.

Yes, we will add MEX/ELS counts plot for bow shock identification.

> Section 3.1.2 : as I wrote above, I would like to see here some evaluation of total oxygen escape rate together with particle escape rate.

See our reply to the first comment above.

> Lines 161: "Assuming that the kinetic energy is constant." This sentence has no main verb.

Yes, it should belong to the sentence before. We will change it to "...square root of mass, assuming that the kinetic energy is constant."

> Line 208 "lighter electrons": delete "lighter", it is not needed

This suggestion will be implemented in a revised paper.

> Figures 6: it is quite hard to see the arrows. Maybe less of them could be shown

This suggestion will be implemented in a revised paper.

> Appendix. I think that adding an appendix just for a table that is referred in the main text is not the best choice. It is better to include the table in the main text.

Yes, we will move it to the main text.